# Genetic and Histopathological Heterogeneity of Neuroblastoma and Precision Therapeutic Approaches for Extremely Unfavorable Histology Subgroups

**DOI:** 10.3390/biom12010079

**Published:** 2022-01-05

**Authors:** Hiroyuki Shimada, Naohiko Ikegaki

**Affiliations:** 1Department of Pathology and Pediatrics, Stanford University School of Medicine, Stanford, CA 94305, USA; 2Department of Anatomy and Cell Biology, College of Medicine, University of Illinois at Chicago, Chicago, IL 60612, USA; ikegaki@uic.edu

**Keywords:** neuroblastoma, International Neuroblastoma Pathology Classification, extremely unfavorable histology, MYC, TERT, ALT

## Abstract

Peripheral neuroblastic tumors (neuroblastoma, ganglioneuroblastoma and ganglioneuroma) are heterogeneous and their diverse and wide range of clinical behaviors (spontaneous regression, tumor maturation and aggressive progression) are closely associated with genetic/molecular properties of the individual tumors. The International Neuroblastoma Pathology Classification, a biologically relevant and prognostically significant morphology classification distinguishing the favorable histology (FH) and unfavorable histology (UH) groups in this disease, predicts survival probabilities of the patients with the highest hazard ratio. The recent advance of neuroblastoma research with precision medicine approaches demonstrates that tumors in the UH group are also heterogeneous and four distinct subgroups—MYC, TERT, ALT and null—are identified. Among them, the first three subgroups are collectively named extremely unfavorable histology (EUH) tumors because of their highly aggressive clinical behavior. As indicated by their names, these EUH tumors are individually defined by their potential targets detected molecularly and immunohistochemically, such as MYC-family protein overexpression, TERT overexpression and ATRX (or DAXX) loss. In the latter half on this paper, the current status of therapeutic targeting of these EUH tumors is discussed for the future development of effective treatments of the patients.

## 1. Introduction

The word “neuroblastoma” is often used as an omnibus term for all peripheral neuroblastic tumors (pNTs), including neuroblastoma, ganglioneuroblastoma and ganglioneuroma. As well defined by Willis [1], neuroblastoma is an embryonal tumor of neural crest origin. We believe that all ganglioneuromas were once neuroblastomas in their early stage of tumor development. Tumors in this group are known to present with a wide range of clinical behaviors, such as spontaneous regression, tumor maturation and aggressive progression refractory to intensive treatment. Because of the difficulty in predicting clinical outcomes of the patients, neuroblastoma was described as an enigmatic disease for many years.

Now, we understand that pNTs do not make a single disease entity; they are heterogeneous and their diverse clinical behaviors are closely associated with genetic/molecular properties of the individual tumors [2,3,4]. In this paper, we briefly summarize prognostic factors and risk classification system of this disease for practical clinical use. Then, the morphological and molecular characteristics of pNTs are outlined for distinguishing tumors in the favorable histology (FH) group and unfavorable histology (UH) group according to the International Neuroblastoma Pathology Classification (INPC). We further detail four subgroups within the UH group, i.e., the MYC subgroup, TERT (Telomerase Reverse Transcriptase), subgroup, ALT (Alternative Lengthening of Telomeres) subgroup and null subgroup. Finally, we discuss precision medicine approaches for the extremely unfavorable histology groups (MYC, TERT and ALT subgroups).

## 2. Prognostic Factors and Risk Classification

In order to predict a wide range of clinical behaviors of the patients with pNTs, there have been several risk grouping systems historically proposed by different cooperative groups, such as the International Society for Paediatric Oncology European Neuroblastoma (SIOPEN), the Gesellschaft für Paediatrische Onkologie und Haematologie (GPOH) and the Children’s Oncology Group (COG). Currently, two systems, the International Neuroblastoma Risk Group (INRG) and the COG risk classification system, are widely accepted; both are based on a combination of so-called prognostic factors, such as clinical stage, patient age at diagnosis, histopathology, and genetic and genomic/molecular properties. The INRG has been designed to facilitate the comparison of risk-based clinical trials conducted in different regions and countries and distinguishes very low (>85% survival), low (75–85% survival), intermediate (50–75% survival) and high-risk (<50% survival) patients [5], while the COG risk classification system has been used for the purpose of patient stratification and protocol assignment in international clinical trials [6,7]. Prognostic factors utilized in the latest and revised COG system include clinical staging, according to the INRG Staging System (INRGSS; Stages L1, L2, M and MS); age at diagnosis (new cut-off points at 365 days, 548 days and 5 years); histopathology, according to the INPC (favorable histology (FH) vs. unfavorable histology (UH)); *MYCN* oncogene status (non-amplified vs. amplified); DNA index (hyperdiploid vs. diploid); and segmental chromosomal aberrations (1p^−^, 11q^−^, 17q^+^; No vs. Yes) [8]. The hazard ratio by each prognostic factor is shown in the Figure 1 and the recent and revised COG neuroblastoma risk classification system is summarized in Figure 2 [8]. Treatment of patients in the low-risk group is often followed by observation (with no chemotherapy or radiation) after surgery or even biopsy alone. Intermediate-risk patients are treated with surgery or biopsy, followed by non-aggressive chemotherapy. High-risk patients are treated with intensive chemotherapy, surgery, radiation therapy, bone marrow/hematopoietic stem cell transplantation and biological-based therapies. The reported 5-year survival rates for non-high-risk and high-risk patients are around 90% and 50%, respectively [8].

## 3. Histopathology of the Peripheral Neuroblastic Tumors

As summarized in Figure 3, according to the INPC, pNTs include four categories and the presence or absence of Schwannian stromal development is noted in parenthesis after the specific tumor category. They are *Neuroblastoma (Schwannian stroma-poor); Ganglioneuroblastoma, Intermixed (Schwannian stroma-rich); Ganglioneuroma (Schwannian stroma-dominant); and Ganglioneuroblastoma, Nodular (composite, Schwannian stroma-rich/stroma-dominant and stroma-poor)* [9]. The *Neuroblastoma* category contains three subtypes—undifferentiated, poorly differentiated and differentiating—based on the grade of neuroblastic differentiation. The *Ganglioneuroma* category contains two subtypes—maturing and mature. The INPC also includes three classes of mitosis–karyorrhexis index (MKI)—low (<100/5000 cells), intermediate (100–200/5000 cells) and high (>200/5000 cells)—as another morphologic indicator for tumors in the *Neuroblastoma* category. As summarized in Table 1, the INPC distinguishes two prognostic groups, the FH group and UH group, based on the age-dependent evaluation of histologic findings [10,11].

### 3.1. Favorable Histology (FH) Tumors

Clinically favorable pNTs seem to show spontaneous regression or age-appropriate tumor differentiation/maturation. The best-known example of spontaneous regression is observed in stage 4S or MS disease whose tumor is classified into the FH group (typically *Neuroblastoma*, poorly differentiated subtype, with a low MKI diagnosed less than 365 days (stage 4S) or 548 days (stage MS)) [12,13]. In the stage 4S or MS disease, neuroblastoma cells characteristically become apoptotic and disappear without differentiation/maturation. While other FH tumors show differentiation/maturation within the age–appropriate framework, tumor differentiation/maturation is prompted by the “cross-talk” supported by critical signaling pathways, such as trkA/NGF signaling and Nrg1/ErbB signaling, between neuronal tumor cells and Schwannian stromal cells [14,15,16]. FH tumors with potential of differentiation/maturation in the *Neuroblastoma* category are poorly differentiated subtype (diagnosed up to 548 days of age at diagnosis) and differentiated subtype (diagnosed up to 5 years of age at diagnosis). All tumors in the *Ganglioneuroblastoma, Intermixed* and *Ganglioneuroma* categories (usually diagnosed in older children even over 5 years of age at diagnosis) are classified into the FH tumors. The prognostic effects by MKI are also age-dependent [17]; a low MKI less than 5 years of age and an intermediate MKI less than 365 days of age indicate favorable clinical outcome.

It is noted that vast majority of the FH tumors are clinically detected and diagnosed right after birth and during the first year of life (Figure 4). Some tumors are found perinatally by ultra-sound test. This could most likely be attributable to the growth rate/speed of the tumors in the FH group. In other words, FH tumors seem to grow rapidly in utero and/or during infancy and then stop growing or slow down the growth toward either spontaneous regression or tumor differentiation/maturation. For example, tumor causing giant hepatomegaly in stage 4S or MS disease is often composed of numerous Ki-67 positive neuroblasts (please see Figure 3G) before entering the phase of spontaneous regression. The numbers of Ki-67-positive cells of FH tumors in 4S/MS disease are more abundant than those of UH tumors in stage 4/M disease, suggesting the growth rate of the former tumors is more rapid than that of the latter tumors. It is also interesting to note that mass-screening systems for neuroblastoma, which started in Japan and preclinically detected neuroblastoma patients by checking urinary VMA/HVA levels at the age of 6 months [18], ended up collecting more than double newly diagnosed cases without significantly changing population-based mortality rates [19]. Those preclinically detected neuroblastomas were almost exclusively favorable, biologically and histopathologically [20,21]. In other words, during the perinatal/infantile period, there seems to be a huge number of spontaneously regressing FH neuroblastomas that would never reach the level of clinical detection.

### 3.2. Unfavorable Histology (UH) Tumors

The vast majority of high-risk cases is composed of tumors in the UH group. They do not seem to show a potential of age-appropriate differentiation/maturation. Among the *Neuroblastoma* category, UH tumors include the undifferentiated subtype (at any age), poorly differentiated subtype (over 548 days of age at diagnosis) and differentiating subtype (over 5 years of age at diagnosis). Adverse prognostic effects of the MKI are also age-dependent—high MKI at any age, intermediate MKI over 548 days and even low MKI over 5 years of age allocate the given tumors to the UH group, respectively [17]. It is reported that *MYCN*-amplified neuroblastoma characteristically presents with high MKI of either undifferentiated or poorly differentiated subtype in the *Neuroblastoma* category (please see Figure 3H) [17,22]. Tumors in the *Ganglioneuroblastoma, nodular* category are composed of distinct components representing different clones. One shows histologic appearance of either *Ganglioneuroma* or *Ganglioneuroblastoma, intermixed* and the other is *Neuroblastoma* characteristically forming a hemorrhagic nodule with or without necrosis. The prognostic distinction of *Ganglioneuroblastoma, Nodular* is based on the same age-linked evaluation of grade of neuroblastic differentiation (subtype) and MKI of *Neuroblastoma* nodule of each case. It is reported that 83% of the patients with *Ganglioneuroblastoma, Nodular* were over 548 days of age at diagnosis and 83% of their tumors were classified into the UH Group [23].

As shown in Figure 4, the tumors in the UH group are clinically detected and diagnosed in children who are older than those with the FH tumors. We speculate that UH tumors seem to grow rather slowly in the utero and during the first year of life compared with the FH tumors and that the UH tumors keep increasing their size with or without metastatic spread, finally becoming clinically detectable with a peak incidence around 2–3 years of age.

## 4. Molecular Heterogeneity of Unfavorable Histology (UH) Neuroblastomas

As mentioned above, the INPC offers the prognostic distinction between the FH group (5-year EFS+/−SE of 88.1+/−0.9; 5-year OS+/−SE of 96.4+/− 0.5) and UH group (5-year EFS+/−SE of 54.6+/−1.3; 5-year OS+/−SE of 65.6+/−1.3) [8]. It is also noted that, along with clinical staging (metastatic disease (Stage M) vs. localized disease (Stages L1, L2) and stage MS), the INPC distinguishes two prognostic groups (FH vs. UH) with very high hazard ratios (95% CI), i.e., an EFS hazard ratio of 4.41 (from 3.86 to 5.03) and an OS hazard ratio of 10.50 (from 8.44 to 13.04), respectively (see Figure 1) [8]. This clearly indicates that new and innovative approaches are urgently needed for patients with highly therapy-resistant UH tumors. Recent progress of neuroblastoma research also clearly supports that different and distinct molecular mechanisms are associated with therapeutic resistance of these aggressive tumors. In this section, those molecular mechanisms, which could be potential targets underlying therapy resistance, are summarized for future incorporation into the INPC for precision prognosis and therapy stratification. Here, we propose four subgroups of UH tumors based on the expression of signature proteins detectable by molecular and immunohistochemical assays (Table 2, Figure 5). These four subgroups of UH tumors are summarized below, of which the MYC, TERT and ALT subgroups are collectively defined as extremely unfavorable histology (EUH) tumors [24].

### 4.1. MYC Subgroup

This subgroup is characterized by the overexpression of MYCN and MYC proteins. The *MYCN* oncogene is considered as a major oncogenic driver of neuroblastoma. *MYCN* amplification is seen in approximately 20–25% of neuroblastoma cases and about 90% of the amplified tumors overexpress MYCN protein [31]. Our studies show that MYCN protein overexpression, rather than the gene amplification, is critical for determining aggressive tumor behavior [25,32]. As we have previously reported, there are also neuroblastomas overexpressing MYC protein [25]. In contrast to MYCN-overexpressing neuroblastoma, it is extremely rare to encounter gene amplification in the MYC-overexpressing neuroblastoma [33]. In this regard, other molecular mechanisms for higher levels of MYC protein expression independent of the genomic amplification, such as the oncogene activation through enhancer hijacking or focal enhancer amplification and MK2-mediated OCT4 transcriptional activation of the gene, have been reported [34,35]. Tumors overexpressing MYC-family protein (MYCN overexpression detected in ~20% and MYC overexpression detected in ~10% of neuroblastomas of the undifferentiated and poorly differentiated subtypes) are collectively named MYC-driven neuroblastomas [25]. These MYC-driven neuroblastomas are highly aggressive and histologically characterized by nucleolar hypertrophy (from one to a few prominent nucleolar formations) [31,36] associated with higher levels of MYC-family protein expression. MYC-driven tumors include large-cell neuroblastoma that is uniquely characterized by enlarged nuclei with one or a few very prominent nucleoli in relatively clear (euchromatic rich) background [37,38]. Large-cell neuroblastoma is reported to express higher levels of stemness markers [39].

### 4.2. TERT Subgroup

Higher levels of TERT expression without MYCN/MYC protein overexpression characterize this subgroup. In this case, TERT overexpression is likely due to long-range genomic rearrangements, but not to promoter mutations in neuroblastoma [40]. *TERT* promoter hyper-methylation may also be involved in its activating mechanism [41,42]. This possibility should be explored in neuroblastoma further. Of note, TERT is the protein component of telomerase, which also includes TERC (telomerase RNA component). Because of high-levels of telomerase activity, cellular senescence is suppressed, which, consequently, contributes to the malignant phenotype of the tumors in this subgroup.

### 4.3. ALT Subgroup

This subgroup is characterized by the loss of Alpha-thalassemia X-linked intellectual disability (ATRX) expression, which results in a telomere-lengthening phenotype via a mechanism involving the homologous recombination DNA-repair machinery. It has been shown that there is a mutually exclusive relationship of ATRX mutation with the MYC and TERT subgroups [26]. In addition, ATRX loss is almost exclusively seen in patients over 5 years of age [43]. Histologically, the ALT subgroup may exhibit nuclear morphology of the conventional “salt-and-pepper” pattern. This is because one of the normal ATRX functions is to maintain transcriptionally active euchromatin by replacing the conventional Histone H3 with the variant Histone H3 (i.e., H3.3) at intragenic regions of selected genes that could form G-quadruplexes (G4) during transcriptional elongation [44,45]. This ATRX function also requires DAXX protein [46]. Because of this, functional loss of DAXX also leads to the ALT phenotype [47].

Of note, some ALT+ cases may not show apparent ATRX loss. In this regard, Chami et al. studied 133 cases of neuroblastoma by ATRX immunohistochemistry and identified 9 cases (~7%) with partial to total loss of ATRX protein. They detected *ATRX* gene mutations in one of the nine ATRX deficient cases and five of the six cases exhibited the ALT phenotype detectably by telomere FISH. They further studied 29 additional ATRX protein-positive cases from patients ≥5 years by telomere FISH and found that six cases were ALT+ (~21%) [48]. Given that, for precise detection of ALT+ cases, null subgroup cases (see below) from patients ≥5 years would require additional screening by ALT-associated PML body (APB) [28], rDNA [29] or telomeric FISH assays [30].

### 4.4. Null Subgroup

There are additional UH neuroblastomas, which do not exhibit MYC-family protein overexpression, TERT overexpression and ATRX (or DAXX) loss. The patients in this null subgroup are anticipated to respond to the current high-risk treatment regimens as long as no other aggravating factors are found [49].

## 5. Therapeutic Targeting of Extremely Unfavorable Histology Neuroblastomas

In this section, we discuss the contribution of pathology to the identification of new precision therapeutic options that could be available to patients with tumors in the EUH subgroup.

### 5.1. Targeting MYC-Family Protein Overexpression

MYC-family proteins are notoriously “undruggable” [50]. They are transcriptional factors that do not have intrinsic enzymatic activities or well-characterized ligand binding sites. In addition, the transactivation domains of MYC proteins are “amorphous” or “disordered” as they do not have stable structure. This hampers crystallographic studies and limits our knowledge on the 3D structure of the MYC transactivation domain (TAD). Together, these characteristics make direct targeting of MYC proteins with small molecules a highly formidable task [50]. Nonetheless, attempts have been made to tackle this hard task.

#### 5.1.1. Direct Targeting Approaches

**MYC/MAX dimerization inhibitors:** MYC-family proteins heterodimerize with the partner, MAX, and become functional molecular complexes as transcription factors. They bind to E-Box sequences throughout the genome and globally modulate the cellular transcriptional activity. One of the ways that inhibits the function of MYC-family proteins is to interfere in the interaction between MYC-family proteins and MAX protein via small molecules. Berg et al. published the first study on MYC/MAX dimerization inhibitors in 2002 [51] and several others have followed this approach since then [52,53,54,55,56,57]. Although the efficacy of these small molecules has been improving, it appears that further refinements are still needed for MYC/MAX dimerization inhibitors before they are given further consideration for human clinical trials.**Strategy to develop small molecules targeting the MYC transactivation domain:** Historically, the direct targeting of transcription factors (TFs) with small molecules has been difficult. However, limited successes have been reported for the TFs that carry ligand-binding sites [58] or have known protein interaction interfaces, such as SH2 domains [59]. In addition, one exceptional case has been reported for small molecules inhibitors of p53–MDM2 interactions [60]. Here, we use the knowledge gained on the interactions between the transcription factor p53 and its effector molecule MDM2 to explore strategies identifying small molecules directly targeting MYC proteins. The N-terminal domain of p53 is essential for its ability to activate gene transcription; thus, it is called transactivation domain (TAD). The p53 TAD is intrinsically disordered and lacks stable tertiary and/or secondary structures under physiological conditions [61]. MDM2 is among the molecules that interact with p53. Genetic and biochemical approaches map the MDM2–p53 interaction sites to the 106-amino acid-long N terminal domain of MDM2 and the N-terminus of the transactivation domain of p53 [62,63]. High-resolution crystal structures of the N-terminal domain of MDM2, complexed with short peptides derived from the N-terminal transactivation domain of p53 (residues 15–29), demonstrated the precise structural requirements for the MDM2–p53 protein–protein interaction [64]. The interaction between p53 and MDM2 involves four key hydrophobic residues in a short amphipathic helix formed by p53 and a small but deep hydrophobic pocket in MDM2 [65]. The existence of such a well-defined pocket on the MDM2 molecule raised the expectation that compounds with low molecular weights could be found that would block the interaction of MDM2 with p53 [64]. In fact, Vassilev et al. reported potent (IC_50_ < 1 μM) and selective small-molecule antagonists of the MDM2–p53 interaction (i.e., Nutlins) and confirmed their mode of action through the crystal structures of the complexes [60]. The apparent reason as to why this approach was successful was that the focus of the study was placed on the structure of p53 effector molecules rather than p53 itself. In this case, it was MDM2, whose 3D structure was determined by X-ray crystallography. By analogy, if one could define a short peptide(s), which could form a stable complex with one of the MYC interacting effector molecules, such as GNC5 [66,67], then one might be able to discover small molecules that interfere in the binding of the MYC peptides (therefore, of MYC proteins) with the MYC effector molecule.

#### 5.1.2. Indirect Targeting Approaches

Given the current status of direct MYC-family protein inhibitors, many have sought indirect approaches to down regulate MYC-family protein expression in cancer cells. The strategies under current consideration include, but are not limited to, (i) transcriptional repression of *MYC* and *MYCN* genes by G4 stabilizers [68,69], by BET (Bromo- and Extra-Terminal)-inhibitors [70,71] and by inhibitors of a transcriptional cyclin-dependent kinase 7 (CDK7) [72,73]; (ii) destabilization of MYCN by Aurora kinase A inhibitors [74] and destabilization of MYC and MYCN proteins by RAS signaling pathway inhibitors [75,76]; and (iii) translational blockade of *MYC* mRNA by eIF4F inhibition and stabilization of RNA G4 by RNA G4 ligands [77,78]. These approaches are discussed below.

**DNA G-*****quadruplex*****stabilizers:** Nucleic acid (DNA and RNA) sequences rich in guanine can form four-stranded structures, called G-quadruplexes (G4) that inhibit transcription. Siddiqui-Jain et al., in 2002, first showed that a G4 structure is present in a promoter region of *MYC* and could be targeted with small G4 stabilizer molecules to repress *MYC* transcription [79]. Interestingly, the *MYCN* promoter also possesses G4 sequences [68], suggesting that G4 stabilizers can target both *MYC* and *MYCN*. To date, several more potent G4 stabilizers have been reported [80,81,82,83,84], including the synthetic fluoroquinolone, Quarfloxin (CX-3543), which was developed for its ability to interact with the *MYC* G4 sequences [69]. However, it turns out that CX-3543 is preferentially concentrated in the nucleoli of cancer cells and inhibits rRNA transcription by RNA Pol I, not at the *MYC* locus [85]. This selective inhibition of Pol I is likely due to the binding of CX-3543 to G4 sequences in the rDNA, as the study identified 14 putative G4-forming sequences on the non-template strand of rDNA [85]. Based on these findings, CX-3543 went into human cancer clinical trials (NCT00955292). Conversely, another fluoroquinolone, CX-5461, which was originally developed as an rRNA synthesis inhibitor (see below), has been shown to exhibit a G4-stabilizing activity [86]. APTO-253 is another small molecule tested in a Phase 1 clinical trial (NCT02267863) and both monomeric APTO-253 and a ferrous complex Fe(253)_3_ stabilize G4 structures present in telomeres, *MYC* and *KIT* promoters [87].**BET (Bromo- and Extra-Terminal)-inhibitors:** BET-family proteins include BRD2, BRD3, BRD4 and BRDT. These proteins bind to acetylated lysine residues of target proteins via their bromodomains. In particular, BRD4 has been shown to accumulate at super-enhancer regions through its binding to acetylated histones [88]. BRD4 co-localizes with an active enhancer marker, histone H3K27Ac, suggesting that BRD4 could directly interacts with H3K27Ac [89]. BRD4 also interacts with the multi-component scaffold protein MEDIATOR and the positive transcription elongation complex P-TEFb [90] to activate gene transcription elongation. Notably, the expression of *MYC* and *MYCN* is under the control of super-enhancers [73,89] and the MYC super-enhancers appear to be essential for MYC-dependent tumorigenesis [91]. The involvement of BRD4 in super-enhancer-driven transcriptional overdrive of *MYC* and *MYCN* expression suggests that BRD4 inhibition with small molecules can be an effective way to down-regulate *MYC* and *MYCN* transcription. Delmore et al. reported, in 2011, the first of its kind, called JQ1; the treatment of MYC-driven cancer cells with JQ1 resulted in rapid down-regulation of MYC protein and its transcriptional network [70]. Subsequently, it has also been shown that JQ1 is effective in down-regulating *MYCN* [71]. Clinically more relevant BET inhibitors have been tested in several clinical trials of adult cancers (NCT02391480 (Drug: ABBV-075), NCT03068351 (Drug: RO6870810), NCT02158858 (Drug: CPI-0610) and NCT02419417 (Drug: BMS-986158)) and OTX015 (NCT02698189), which has been shown to be effective against mouse and human MYCN-driven tumor models [92].**Inhibitors of Cyclin-dependent kinases 7:** The active involvement of super-enhancers of the over-driven *MYCN* expression in *MYCN*-amplified neuroblastoma came to light through the study of a covalent CDK7 inhibitor, THZ1 [73]. CDK7 binds to cyclin H and MAT1 to form a Cdk-activating kinase (CAK). CAK is a component of the transcription factor TFIIH, which is involved in transcription initiation [93]. TFIIH is a component of general transcription factors and recruits RNA Pol II to the promoters of genes to complete the formation of the transcription pre-initiation complex (PIC) [94]. Phosphorylation by CDK7 of the C-terminal domain (CTD) of serine at position 5 (S5) of RNA Pol II initiates gene transcription [94]. Subsequently, Pol II advances to the pause site, where it is stabilized by pausing factors, such as negative elongation factor (NELF) and DRB-sensitivity-inducing factor (DSIF). P-TEFb is the main factor required to release paused Pol II from the promoter-proximal region. P-TEFb is composed of CDK9 and cyclin T. P-TEFb phosphorylates the CTD of serine at position 2 (S2). Phosphorylation of the CTD at the S2 position fully activates transcription elongation [95]. Super-enhancers, together with gene specific transcription factors, BRD4 and the MEDIATOR complex, help assemble PIC more efficiently and initiate transcription via TFIIH/CAK (the CDK7 complex) at the transcriptional start site. Then, super-enhancer-bound BRD4 and the super elongation complex (SEC) recruit P-TEFb to release paused Pol II via phosphorylation of CTD at S5 [95]. Because of the hyper-activated super-enhancer-driven transcription of *MYCN*, the CDK7 inhibitor, THZ1, preferentially suppresses *MYCN* transcription. A similar finding was made for *MYC* in response to THZ1 in *MYC*-amplified SCLC (small cell lung cancer) cells [72]. Surprisingly, inhibition of CDK9 by a small molecule inhibitor, i-CDK9, enhances MYC expression and only simultaneous inhibition of CDK9 and BRD4 can efficiently induce growth arrest and apoptosis in cancer cells [96]. More clinically suitable CDK7-inhibiting drugs, SY-1365 and CT7001, are tested in Phase I clinical trials for advanced solid tumors (NCT03134638 and NCT03363893, respectively).**Destabilization of MYCN by Aurora kinase A inhibitors:** Otto et al. revealed, in 2009, that Aurora A binds to and stabilizes MYCN by preventing its FBXW7 E3 ubiquitin ligase-mediated proteasomal degradation [97]. This activity is independent of Aurora A’s catalytic activity. Gustafson et al. have further shown that inhibitors disrupting the native conformation of Aurora kinase A cause the degradation of MYCN protein [74]. Alisertib (MNL8237), an investigational inhibitor of Aurora kinase A developed by Takeda Inc., currently being tested in several adult cancer clinical trials, binds to the catalytic domain of Aurora kinase A and causes an allosteric change in the protein structure. This, in turn, releases MYCN from Aurora kinase A and causes proteasome-mediated MYCN degradation [98]. Alisertib had been tested in a Phase I trial by COG [99] and is also tested in combination with Irinotecan and Temozolomide (NCT01601535). However, Takeda terminated an Alisertib Phase 3 trial relapsed or refractory peripheral T-cell lymphoma in adult and its future in pediatric cancer trials is uncertain [100].**Destabilization of MYC/MYCN by inhibitors of RAS signaling pathways:** Two arms of RAS signaling pathways regulate the stability of MYC-family proteins through phosphorylation at T58 and S62 of MYC or equivalent residues of other MYC family members. The phosphorylation at S62 by ERK of the MEK/ERK arm of RAS pathways enhances MYC protein stability; this phosphorylation subsequently allows T58 to be phosphorylated by active (non-phosphorylated) GSK3β, whereas the PI3K/PDK1/AKT arm of RAS pathways inactivates GSK3β by phosphorylation; when MYC proteins are doubly phosphorylated at T58 and S62, the protein is recognized by the protein phosphatase PP2A, which keeps GSK3β active and dephosphorylates S62. T58 monophosphorylated MYC proteins are then recognized by FBWX7 E3 ubiquitin ligase, which *polyubiquitinates* MYC proteins for proteasomal degradation. Therefore, inhibition of either arm of RAS signaling pathways results in destabilization of MYC-family proteins. Over the years, potent and selective inhibitors of RAS signaling pathways have been developed and tested in human clinical trials [101,102,103]. Some of the RAS pathway inhibitors are tested in The Pediatric MATCH Screening Trial (NCT03155620).**Translational initiation blockade of MYC mRNA by eIF4F inhibition**: One of the components of the eukaryotic translation initiation complex (EIF) is eIF4A, which functions as an ATP-dependent DEAD-box RNA helicase to remove secondary structures from 5′UTR of mRNA. Ribosome-profiling assay experiments have shown that eIF4A regulates mRNA translation of transcripts with 5′UTRs containing G4-forming CGG motifs. Among the most eIF4A-sensitive mRNAs, there are several oncogenes and transcription factors (e.g., MYC, MYB, NOTCH, CDK6 and BCL2) [78]. These RNA G4 sequences may be selectively targeted by RNA G4 specific ligands [104]. These observations suggest that inhibition of eIF4A and/or stabilization of RNA G4 at 5′UTR of mRNA can suppress translation of the eIF4A-sensitive mRNAs. In fact, an inhibitor of eIF4A, Silvestrol, causes marked down-regulation of MYC, MYB, NOTCH, CDK6 and BCL2 proteins [78]. It has also been reported that Silvestrol inhibits MYC expression and suppresses tumor growth in vivo [105]. There are several additional small molecules with the eIF4A inhibiting activity, including Rocaglates, a class of natural products derived from plants of the Aglaia genus [106,107]. Rocaglamide has been shown to downregulate MYC expression [108]. CR-1-31B (CR-31) is a synthetic rocaglate [106]; when treated with CR-1-31-B at low nanomolar doses (≤20 nM), neuroblastoma cell lines exhibit decreased viability and increased apoptosis rates, as well as changes in cell cycle distribution [109]. These could be attributed to down-regulation of MYC and MYCN by CR-1-31-B. Finally, eFT226 is a clinically more relevant small molecule that inhibits translation of specific mRNAs, including MYC. Consequently, it exhibits potent antiproliferative activity and significant in vivo efficacy against a panel of diffuse large B-cell lymphoma (DLBCL) and Burkitt lymphoma mouse models with ≤1 mg/kg/week intravenous administration [110]. Moreover, the results of a Phase 1–2 study of eFT226 in advanced solid tumor malignancies are also encouraging (NCT04092673) [111]. Conversely, other translational initiation inhibitors have been tested in clinical trials, including ribavirin (e.g., NCT02073838) and eFT508 (e.g., NCT02605083). Ribavirin, a broad-spectrum antiviral drug, mimics the m7G cap structure, whereas eFT508 is an inhibitor of mitogen-activated protein kinase interacting kinase 1 and 2. Both drugs target eIF4E to inhibit translation initiation.**Targeting MYC-driven hypertrophic nucleoli:** As stated earlier, MYC-driven neuroblastomas characteristically show prominent nucleolar formation, a sign of increase in ribosome synthesis and translation. Therefore, inhibition of rRNA gene transcription and protein translation by small molecule inhibitors could represent potential therapeutic approaches for MYC-driven neuroblastomas [31]. In fact, we have explored the efficacy of small molecule inhibitors of rRNA synthesis and protein translation, such as CX-5461 and Halofuginone, respectively [31,112] (see below). These small molecules are clinically relevant as they have been tested for their efficacy in various human diseases in clinical trials [113,114].

CX-5461 inhibits the binding of the SL1 transcription factor to the rDNA promoter and prevents the initiation of rRNA synthesis by RNA Pol I [115]. Consequently, CX-5461 halts ribosomal assembly, resulting in the accumulation of unassembled ribosomal proteins. Then, free ribosomal proteins promote cancer-specific activation of p53 [113]. Moreover, a similar set of free ribosomal proteins could also down-regulate MYC protein by distinct mechanisms [116,117]. Halofuginone inhibits glutamyl-prolyl tRNA synthetase [118], leading to the accumulation of uncharged prolyl tRNAs. Then, the accumulated prolyl tRNAs cause suppression of protein translation via the induction of amino acid starvation response [119]. In addition, inhibition of glutamyl-prolyl tRNA synthetase may free one of the cofactors (AIMP2-Aminoacyl TRNA Synthetase Complex Interacting Multifunctional Protein 2) from the human multi-tRNA–synthetase complex. Then, freed AIMP2 could migrate into the nucleus and inhibit FBP (FUBP1). FBP is the transcriptional activator of *MYC* [120,121]. As a result, Halofuginone could induce down-regulation of MYC-family proteins through both global translational suppression and transcriptional inhibition of *MYC*. We have shown that treatment of MYC-driven neuroblastoma cells with CX-5461 and Halofuginone causes down-regulation of MYC and MYCN proteins and growth suppression MYC-driven neuroblastoma cell lines in vitro and preclinical models [31,112]. As stated above, the effect of CX-5461 on MYC-family protein expression may also be due to its direct effect on transcription of *MYC* and *MYCN* through the binding to DNA G4 sequences in their promoter regions.

In addition, there are additional ways in which hypertrophic nucleoli can be targeted. Actinomycin D was the first antibiotic used as a treatment of pediatric tumors, including Wilms’ tumor, Rhabdomyosarcoma and Ewing’ s sarcoma [122,123,124]. Actinomycin D is a DNA intercalator with the preference for GC-rich DNA sequences [125]. It preferentially binds to the GC-rich promoter of the 45S ribosomal gene at low concentrations and inhibits RNA Pol I-dependent transcription, leading to a disruption of ribosome biogenesis [125]. Triptolide is a diterpenoid epoxide, extracted from the thunder god vine, *Tripterygium wilfordii*. It has been reported that Triptolide interrupts ribosomal rRNA synthesis through inhibition of RNA Pol I and UBF transcriptional activation, leading to nucleolar disintegration [126].

### 5.2. Targeting Telomere Maintenance and Elongation

#### 5.2.1. TERT Inhibitors

For those patients with TERT-overexpressing neuroblastoma, clinical trials with telomerase inhibitors can be considered. Imetelstat (GRN163L; JNJ-63935937) [127] is a potent and specific inhibitor of telomerase that binds with high affinity to TERC. Imetelstat is administered by intravenous infusion and it has been tested in human clinical Phase 2 and 2/3 trials [127]. However, as pointed out by Guterres and Villanueva [128], although Imetelstat has elicited robust response rates in phase II trials involving patients with myelofibrosis or essential thrombocytopenia [129,130], the mechanism of action of Imetelstat in responders may be due to sequence-independent side effects of phosphoramidates on immunostimulation and not due to telomerase inhibition [131], as responses neither correlated with baseline telomere length, nor was telomere shortening observed in responders.

Interestingly, it has been shown that Sorafenib and its derivative UC2288 synergize with Imetelstat to inhibit growth of mouse xenografts of human cancer. Sorafenib inhibits several tyrosine protein kinases, including VEGFR, PDGFR and Raf-family kinases, and has been tested in combination with topotecan and cyclophosphamide in relapsed neuroblastoma patients by the New Approaches to Neuroblastoma Therapy (NANT) consortium (NCT02298348) [132]. Besides its kinase inhibiting activity, Sorafenib suppresses p21 expression induced by chemotherapy [133]. Furthermore, a Sorafenib derivative, UC2288, with minimal kinase inhibitory and more selective p21 inhibiting activities, also synergizes with Imetelstat to inhibit growth of mouse xenografts of human cancer [134]. These observations suggest that Sorafenib’s ability to synergize with Imetelstat is due to its p21-attenuating activity, but not its kinase-inhibiting activity. Nonetheless, it is expected that a combination of Imetelstat and Sorafenib may be more efficacious against those neuroblastomas with elevated telomerase activity.

In addition, it was reported that a combination therapy of a BET bromodomain inhibitor, OTX015, and a proteasome inhibitor, carfilzomib, synergistically blocked TERT expression, induced tumor cell apoptosis, suppressed tumor progression and improved survival of a preclinical mouse PDX model, which was largely reversed by forced TERT overexpression. Thus, combination therapy would likely be translated into the first clinical trial of a targeted therapy in patients with *TERT*-rearranged neuroblastoma [135]. BIBR1532 is another potent and selective inhibitor of human telomerase [136]. It has been shown that BIBR1532 exerts potential anti-cancer activities in preclinical models of feline oral squamous cell carcinoma through inhibition of telomerase activity and down-regulation of TERT [137].

#### 5.2.2. Inhibitors for ALT Phenotype

ALT inhibition could also be considered in the clinical trials for those patients with neuroblastoma having loss of ATRX. However, because ATRX loss is due to structural alterations in the *ATRX* gene, including in-frame deletions, missense, nonsense and frame-shift single-nucleotide variations (SNVs) [138], it would be difficult to regain the expression of ATRX in neuroblastoma. To address this problem, it is essential to understand normal ATRX functions and the mechanistic of ALT.

ATRX forms an SWI/SNF-like chromatin remodeling complex with DAXX to insert a variant Histone H3, H3.3, into pericentric and telomeric chromatin [139]. This maintains the heterochromatic state in the pericentromeric and telomeric regions. The presence of H3.3 at telomeres correlates with the repression of the telomeric repeat-containing RNA (*TERRA*) expression, a long non-coding RNA [140]. In non-ALT cells, RPA at telomeres are replaced by POT1 (protection of telomeres 1) because of the decline in *TERRA* during G2/M via the hnRNPA1-mediated mechanism [140]. During the S phase, *TERRA* is very abundant and associates with telomeres. In fact, *TERRA* can sequester hnRNPA1 and prevent it from binding to DNA. During the S phase, RPA can bind single-strand telomere DNA very efficiently to facilitate DNA replication, but, interestingly, the level of *TERRA* starts declining after DNA replication in the G2/M phase. ATR (Ataxia Telangiectasia and Rad3-related protein) activation by RPA–ssDNA is inhibited by POT1 recruitment at telomeres during G2/M. ATRX loss progressively induces chromatin de-compaction at telomeres [141], which is considered prerequisite for the ALT phenotype. In ALT cells, loss of ATRX also interferes with the decline in *TERRA* in the G2/M phase. This leads to persistent association of replication protein A (RPA) with telomeres after DNA replication. Consequently, a recombinogenic nucleoprotein structure (RPA–ssDNA) at telomeres persists during the G2/M phase. This RPA–ssDNA at telomeres is recognized as a replication error of DNA by ATR along with a partner protein called ATRIP (ATR Interacting Protein). This activates the ATR-medicated DNA damage response. Consequently, this results in the activation of CHK1 (Checkpoint Kinase 1). The activated CHK1, in turn, results in phosphorylation and nuclear translocation of RAD51 recombinase. RAD51 plays a central role in homologous recombination repair of DNA. It catalyzes strand transfer between a broken sequence and its undamaged homologue to allow re-synthesis of the damaged region to happen. RPA-coated single strand DNA is a key intermediate for homologous recombination and ALT is a homologous recombination-like process, suggesting that loss of ATRX gives rise to a recombinant genic structure at telomeres, thus promoting ALT pathway activation. BRCA2 (Breast Cancer 2) is also involved in the process to remove RPA from ssDNA and replace it with RAD51. In ALT cells, this mechanism appears to be used to maintain telomeres [142].

Taken together, ATR likely plays a key role in maintenance of ALT. Interestingly, it has been shown that inhibition of ATR disrupts ALT and triggers chromosome fragmentation and apoptosis in ALT cells. Moreover, cell death induced by ATR inhibitors appears highly selective for ALT cells [142]. There are a number of clinically relevant ATR inhibitors that are now available. For example, BAY1895344 is a potent and orally bioavailable ATR inhibitor and results of the first human clinical trial on BAY1895344 (NCT03188965) have been reported with a favorable outlook [143]. AZD6738 is another orally bioavailable and selective ATR kinase inhibitor and is currently being tested in several Phase I and II clinical trials in adults [144]. As mentioned, CHK1 comes downstream of ATR and facilitates homologous recombination repair (HRR). Thus, CHK1 inhibitors, such as MK8776/SCH900776 [145], could also be selectively efficacious against ATL cells.

Of note, several studies have further suggested that not all ALT+ cancer cell lines, including neuroblastoma lines, exhibited hypersensitivity to ATR inhibitors [146,147]. Subsequently, it was reported that ALT+ neuroblastoma cell lines exhibit activation of the ATM pathway, rather than the ATR pathway, and that an ATM inhibitor reverses their chemoresistance against temozolomide + SN-38 (the active metabolite of irinotecan) [148]. Thus, it remains to be seen as to what mechanisms are involved in the initiation and maintenance of the ALT phenotype in cancer cells. However, it is plausible that the initiation phase of ALT may predominantly involve the ATR pathway and the maintenance phase of ALT may principally require additional ATM pathway activation. If so, a combination of ATM and ATR inhibitors may prove more efficacious against ALT+ neuroblastomas compared to either inhibitor alone.

#### 5.2.3. Induction of Telomere Erosion

Xu et al. have discovered that CX-5461 (and related CX-3543) binds and stabilizes G4 DNA structures in vitro, inhibits the progression of DNA replication complexes and results in increased in vivo G4 structures. They further investigated the effect of CX-5461 on the integrity of telomeres, which are enriched with G4 structures. Telomere FISH results showed an increased frequency of telomere defects in both *BRCA2^+/+^* and *BRCA2^−/−^* HCT116 cells after exposure to CX-5461 and this defect was more prominent in *BRCA2^−/−^* cells. Collectively, these data suggest that CX-5461 is a G4 stabilizer and induces genome instability specifically at G4 sequences [86]. These observations suggest that CX-5461 is preferentially efficacious against neuroblastoma cells with telomere maintenance and elongation aberrations. In addition, ATR inhibitors have been reported to synergize with CX-5461 and the combination exhibits more robust cancer cell killing of leukemia cells [149]. This observation suggests that a combination of ATR inhibitors and CX-5461 may be a better option for the treatment of ALT neuroblastomas.

### 5.3. An Emerging Common Strategy for Therapy-Resistant or -Refractory UH Neuroblastomas

Based on the discussion above, one common strategy to treat therapy-resistant or -refractory UH (EUH) neuroblastomas appears to be DNA G4 stabilizers, such as CX-5461. This drug can target MYC-family-driven, TERT-overexpressing and ALT-phenotype neuroblastomas and it may be considered the core agent in a cocktail of drugs to treat aggressive EUH neuroblastomas.

## 6. Conclusions

For the last 40 years, the pathology classification of peripheral neuroblastic tumors (pNTs) has made a pivotal contribution to prognosis and therapy stratification for children with this disease. Unlike most adult cancer pathology classifications, the Shimada classification system [150] and the subsequent International Neuroblastoma Pathology Classification (INPC) [9,10] incorporated age as the critical factor of prognostic evaluation of histologic indicators, taking into account the age-distribution of the patients and development of the peripheral nervous system during the period of disease occurrence. To date, the INPC can clearly define the favorable histology (FH) and unfavorable histology (UH) groups of pNTs, having over 90% and below 50% survival probabilities, respectively. This makes the INPC the most significant prognostic factor, with the highest hazard ratio distinguishing overall survival rates of the patients in the current COG risk classification scheme. However, this also leaves one important unsolved issue that at least one in two patients with UH tumors succumbs to the disease despite receiving multimodal and high-intensity therapy regiments.

The proposed distinction of UH neuroblastomas into the therapy-responsive “null” subgroup and therapy-resistant/refractory “EUH” subgroups constitutes the first step forward to precision medicine approaches for treating the patients with pNTs effectively [24]. The target-defined subclassification of UH tumors, based on molecular and immunohistochemical assays detecting overexpression of MYC-family proteins and TERT and the loss of ATRX (or DAXX), has a significant merit, because one can selectively identify the drivers of the disease for directing the specific therapeutic approaches. As discussed in this paper, vigorous attempts have been made to develop such therapeutics and the idea of “undruggable” targets by small molecules may be fading away. We hope that, in near future, such therapeutics become available for the effective treatment of EUH neuroblastomas to save the children without severe late side effects.

## Figures and Tables

**Figure 1 biomolecules-12-00079-f001:**
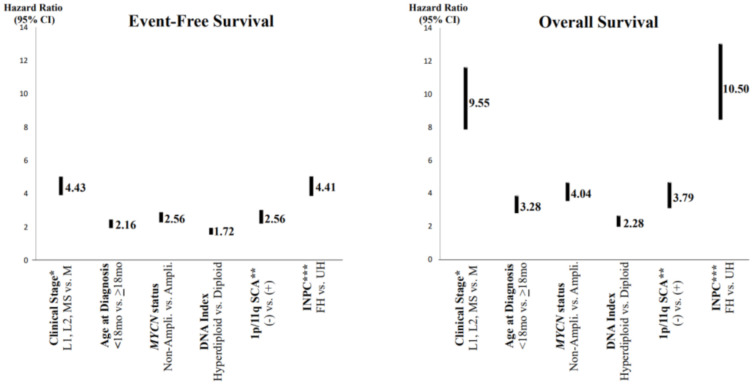
Neuroblastoma, prognostic factors and hazard ratios. * Clinical stage according to the International Neuroblastoma Risk Group Staging System; ** SCA = segmental chromosomal aberrations; *** INPC = International Neuroblastoma Pathology Classification (FH = favorable histology, UH = unfavorable histology). Note: Clinical stage (localized (L1 and L1) and MS (metastatic, special) disease vs. M (metastatic) disease) and INPC (FH group vs. UH group) distinguish tow prognostic groups with very high hazard ratio.

**Figure 2 biomolecules-12-00079-f002:**
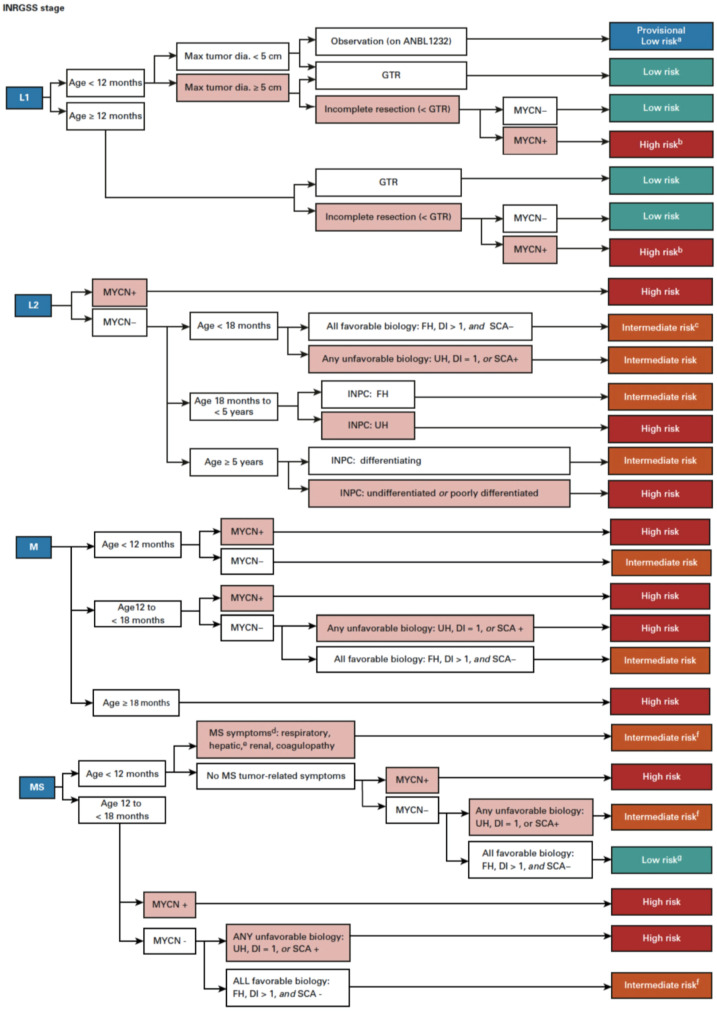
Children’s Oncology Group revised neuroblastoma risk classification system. For detailed explanation of the revised classification system, please see Ref. [8]. (Irwin M. et al., 2021 [8]).

**Figure 3 biomolecules-12-00079-f003:**
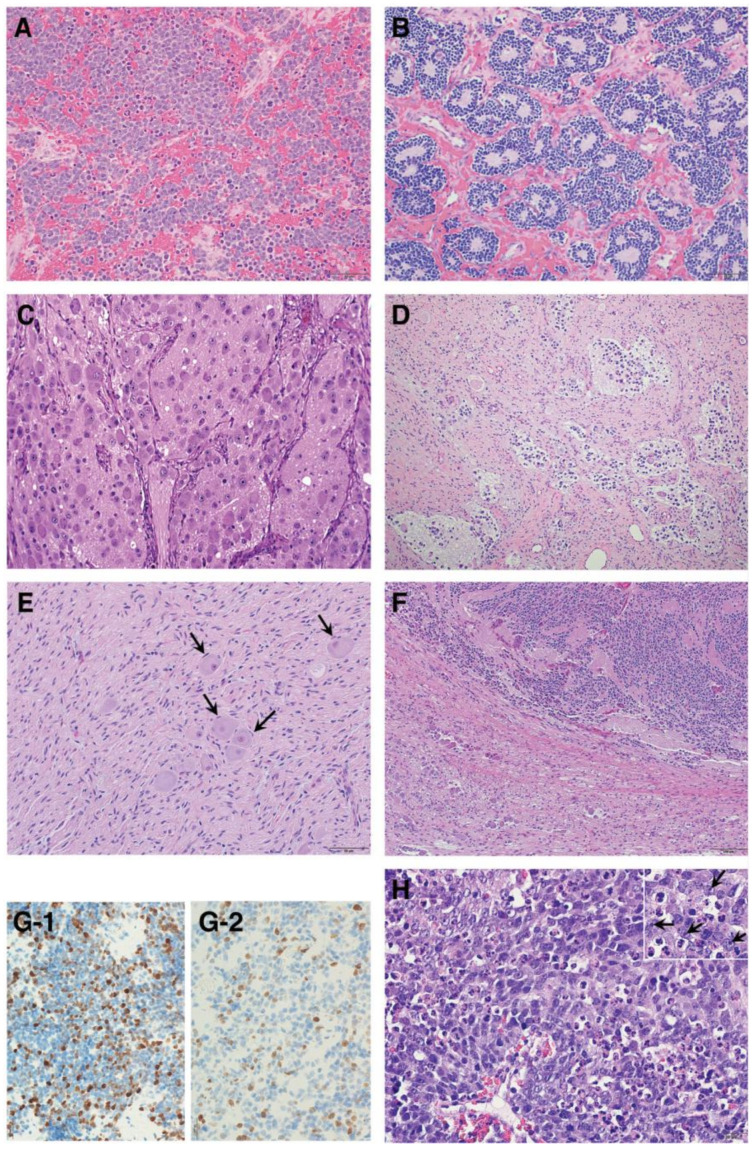
Histologic features of peripheral neuroblastic tumors: (**A**) neuroblastoma (Schwannian stroma-poor), undifferentiated subtype (original, ×200); (**B**) neuroblastoma (Schwannian stroma-poor), poorly differentiated subtype (original, ×200); (**C**) neuroblastoma (Schwannian stroma-poor), differentiating subtype (original, ×200); (**D**) ganglioneuroblastoma, intermixed (Schwannian stroma-rich) (original, ×100); (**E**) ganglioneuroma (Schwannian stroma-dominant)—note that completely mature ganglion cells are covered with satellite cells (arrows) (original, ×200); (**F**) ganglioneuroblastoma, nodular (composite, Schwannian stroma-rich/stroma-dominant and stroma-poor)—note two distinct histologies, i.e., neuroblastoma in the upper half and ganglioneuroma in the lower half in this case (original ×100); (**G**) Ki-67 immunostaining on two neuroblastomas of poorly differentiated subtype with a low mitosis–karyorrhexis index (original ×400). Note: (**G-1**) shows biopsy from rapidly enlarging liver of 3-month-old baby (Stage MS case, favorable histology) before starting spontaneous regression; (**G-2**) shows biopsy from abdominal mass of 5-year-old child (Stage M case, unfavorable histology). The favorable histology tumor contains numerous and more Ki-67 positive nuclei than the unfavorable histology tumor. (**H**) MYCN oncogene-amplified neuroblastoma demonstrating a characteristic histology of poorly differentiated subtype with a high mitosis–karyorrhexis index (original, ×400). Note: Tumor cells show nucleolar hypertrophy (Inset: prominent nucleoli indicated by arrows).

**Figure 4 biomolecules-12-00079-f004:**
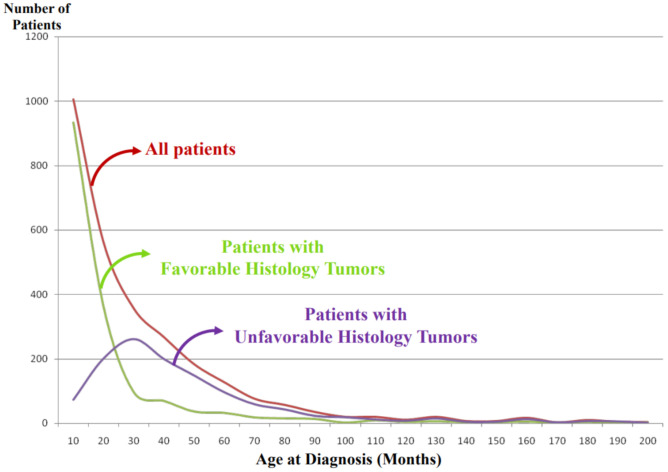
International Neuroblastoma Pathology Classification: patient distribution by age at diagnosis. Note: Most of the patients having favorable histology tumors are clinically detected and diagnosed in younger-age groups than the patients having unfavorable histology tumors.

**Figure 5 biomolecules-12-00079-f005:**
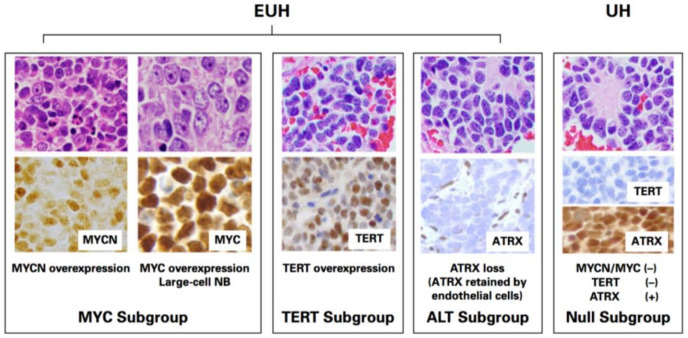
Proposed subgroupings of unfavorable histology neuroblastomas. Four subgroups, i.e., MYC (overexpressing either MYCN or MYC protein), TERT (*TERT* gene activation due to rearrangements), ALT (due to ATRX or DAXX loss) and null (no MYC, No TERT, no ALT), are distinguished immunohistochemically. EUH, extremely unfavorable histology; UH, unfavorable histology. The immunohistochemical grading of MYC-family protein expression was performed according to Ref. [25] using monoclonal anti-MYCN, NCM II 100 and anti-MYC, Y69. The evaluation of ATRX loss was performed when tumor cell nuclei were negative and endothelial cell nuclei (built-in-control) were positive in the same tumor tissue using polyclonal HPA001906. The immunohistochemical grading of TERT was performed essentially the same way as MYC-family proteins [25] using monoclonal A-6. (Adopted from the article by Ikegaki and Shimada, *JCO Precis Oncol* 2019;3:PO18.00312 [24] with the publisher’s permission.). (All images, original ×600).

**Table 1 biomolecules-12-00079-t001:** International Neuroblastoma Pathology Classification.

			Age at Diagnosis		
Neuroblastoma (Schwannian stroma-poor)			<548 days	548 days–5 years	≥5 years
	Undifferentiated subtype				
With		Any MKI			
	Poorly differentiated subtype				
With		Low MKI			
		Intermediate MKI			
		High MKI			
	Differentiated subtype				
With		Low MKI			
		Intermediate MKI			
		High MKI			
Ganglioneuroblastoma, Intermixed			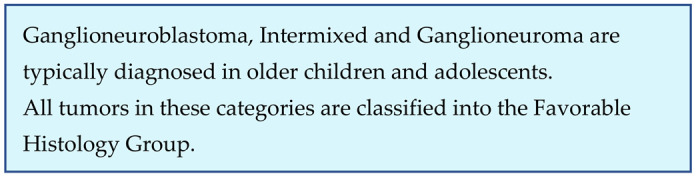		
	(Schwannian stroma-rich)				
Ganglioneuroma					
	(Schwannian stroma-dominant)				
Maturing subtype					
Mature subtype					
Ganglioneuroblastoma, Nodular			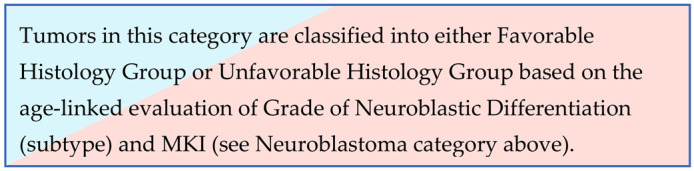		
(Composite, Schwannian stroma-rich/					
stroma-dominant and stroma-poor)					




Blue: Favorable Histology; Red: Unfavorable Histology; MKI = Mitosis-Karyorrhexis Index.

**Table 2 biomolecules-12-00079-t002:** Proposed subgroups of unfavorable histology neuroblastoma.

Subgroups	Immunohistochemistry (IHC) Markers *			Histologic Markers	Predicted Survival on the Current Therapy †
	Pan-MYC or (MYCN/MYC)	TERT	ATRX		
MYC-driven					
MYC	Overexpression	Overexpression or No overexpression	Retention	Nucleolar hypertrophy (Including Large Cell Neuroblastoma)	Dismal
Non MYC-driven					
TERT	No overexpression	Overexpression	Retention	Salt-and pepper nuclei; Conventional neuroblastoma	Dismal
ALT	No overexpression	No overexpression	Loss ‡	Salt-and pepper nuclei; Conventional neuroblastoma	Dismal
Null	No overexpression	No overexpression	Retention	Salt-and pepper nuclei; Conventional neuroblastoma	Better Response

* The immunohistochemical grading of MYC-family protein expression was performed according to Ref. [25]. The evaluation of ATRX loss was performed when tumor cell nuclei were negative and endothelial cell nuclei (built-in-control) were positive in the same tumor tissue. The immunohistochemical grading of TERT was performed essentially the same way as MYC-family proteins [25]. † The estimated survival of each subgroup is based on the references [25,26]. ‡ To exclude the possibility of overlooking ALT cases with no apparent ATRX loss (including ATRX-independent ALT cases) [27] and rare *DAXX* mutations, patients over 5-years of age in the null group can be subjected to additional screening by ALT-associated PML body (APB) [28], rDNA [29], or telomeric FISH assays [30]. (Adopted from the article by Ikegaki and Shimada, *JCO Precis Oncol* 2019;3:PO18.00312 [24] with the publisher’s permission).

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
