# Peer review of "Genetic and Histopathological Heterogeneity of Neuroblastoma and Precision Therapeutic Approaches for Extremely Unfavorable Histology Subgroups"

_biomolecules, 2022, doi:10.3390/biom12010079_

Round 1

Reviewer 1 Report

In the present paper, Hiroyuki Shimada and Naohiko Ikegaki described recent advance of neuroblastoma research with precision medicine approach demonstrates that tumors in the UH Group are also heterogene ous, and 4 distinct subgroups; MYC, TERT, ALT, and Null, are identified. Among them, the first 3 subgroups are collectively named as Extremely Unfavorable Histology (EUH) tumors because of their highly aggressive clinical behavior. As indicated by their names, these EUH tumors are individually defined by their potential targets detected molecularly and immunohistochemically; such as MYC-family protein overexpression, TERT overexpression, and ATRX (or DAXX) loss. This review provides recent insights into our understanding of genetic background of Extremely Unfavorable Histology neuroblastoma and highlights emerging therapeutic strategies as expected treatments for EUH NB tumors.

This is well written, comprehensive, and highly relevant review addressing fundamental pathogenic mechanisms and treatment approaches in Extremely Unfavorable Histology NBs. I think this review is suitable for Biomolecules and would like to ask several questions to authors before publication in Biomolecules.

 Major points

  1. Page 9 of 26, authors presented n-MYC, c-MYC, ATRX, and TERT expression by immunohistochemistry. This Figure is modification of the figure in ref 24. In the ref 24, I could not find the sentences and refs described about the immune-histochemical methods and results of TERT and ATRX immune-staining. Authors had better indicate the appropriate refs for TERT and ATRX immune-staining. Moreover, Rose Chami et al described about an interesting phenomenon about immunohistochemistry for ATRX and ATRX mutations (Am J Surg Pathol, Volume 43, Number 9). Authors should mention this paper and indicate their own findings about IHC and mutation of ATRX.
  2. About TERT expression, MYCN (n-MYC) is known to induce TERT transcription. Authors can mention the TERT expression in the MYC subgroup by immunohistochemistry?
  3. Page 10of 26, 4.2. TERT subgroup, authors described about TERT promoter hypermethylation in refs 36/37, however, ref 36 is about medulloblastoma and relationship between TERT expression and TERT promoter was not written in ref 37.
  4. Recently, a Science Trans Medicine paper “ALT neuroblastoma chemoresistance due to telomere dysfunction–induced ATM activation is reversible with ATM inhibitor AZD0156” was published. This paper is important for this area and authors had better mention in this review.

Minor points:

  1. Page 7 of 26, lines 137-138, [17] can be moved to the last of this sentence?
  2. Page 16 of 26, lines 574-576 are OK?
  3. Page 16 of 26, line 578, What is the “8”?

Reviewer 2 Report

The manuscript is related to neuroblastoma disease and heterogenity and unfavorable risk and evalute the potentail therapeutic targets for this unfavaroble histologic subgrops. The issue is very important for neuroblastoma and treatment.

This review is well designed and discussed almost all concerns related diseases and components. This manuscript is discussing the disease components to researchers for the filling the gap related to unfavaroble  groups of neuroblastoma. This review study is very valuble evalution for unfavorable neuroblastoma disease.

Author Response

No critique to respond.

Reviewer 3 Report

In the manuscript entitled "Genetic and Histopathological Heterogeneity of Neuroblastoma and Precision Therapeutic Approaches for Extremely Unfavorable Histology Subgroups," the authors compile and review in a practical way the published landscape of genetics, histopathology, and therapeutic guidance especially for high-risk neuroblastoma. However, some revisions must be made.

Major review

1.- Some statements are hypotheses, assumptions or personal opinions without references. Could the authors show scientific data to prove the claims on the pages indicated below?

  • Related to Ki-67 positivity: page 6 (lines 114-115), page 7 (lines 144-149) and Fig 3: G-1 and G-2.
  • Related to UH tumor growing: page 8 (lines 181-184).
  • Related to Null subgroup response to treatment regimens: page 10 (lines 260-261).

2.- Since the image in Figure 3B could be confusing with Willms tumor, could the authors present another pdNB?

Minor review:

- There are several scientific terms that are abbreviated multiple times (i.e. pNTS, FH,UH,TERT,ALT,TAD,G4...); others that are not abbreviated from the beginning (i.e. EUH ...) or that after being abbreviated are used again without abbreviation (i.e. G4..).

- Publications from 2013 and 2015 are not recent studies. Please modify the phrase (page 9)

- There are several small spelling mistakes (i.e., no end points of the phrase, or their substitution by colons, or drug names written indistinctly in upper and lower case).       

- There are some typographical errors (explanation of Fig 1, lines: 171, 479 ...).

- The first sentence of section 5.3 need to be checked because is fragmented or cut.

Round 2

Reviewer 1 Report

  1. Page 9 of 26, authors presented n-MYC, c-MYC, ATRX, and TERT expression by immunohistochemistry. This Figure is modification of the figure in ref 24. In the ref 24, I could not find the sentences and refs described about the immune-histochemical methods and results of TERT and ATRX immune-staining. Authors had better indicate the appropriate refs for TERT and ATRX immune-staining.

Response: Due to the copyright issue, we divided Figure 5 of the MS into the original form of Table 1 (Table 2 in the MS) and Figure 1A (Figure 5 of the MS) with a note that they are adopted in this chapter. The legend now describes the IHC method and the associated references.

Reviewer 1: This response is acceptable. However, I wrote my opinion in the next comment.

  1. Moreover, Rose Chami et al described about an interesting phenomenon about immunohistochemistry for ATRX and ATRX mutations (Am J Surg Pathol, Volume 43, Number 9) {Chami, 2019 #1968}. Authors should mention this paper and indicate their own findings about IHC and mutation of ATRX.

Response: The reference mentioned is included in the MS and we discuss their finding in the MS. At this moment, we will continuously perform both IHC and molecular test for ALT, and correlate the results with clinical behavior of neuroblastomas. In this regard, we have modified the section 4.3. ALT subgroup in the text. In addition, we have added a sentence in the Table 2 legend that “To exclude the possibility of overlooking ALT cases with no apparent ATRX loss (including ATRX-independent ALT cases)[27] and rare DAXX mutations, patients over 5-years of age in the Null group can be subjected to additional screening by ALT-associated PML body (APB)[28], rDNA [29] or telomeric FISH assays [30].”

Reviewer 1: Basically, this answer is acceptable.

This reviewer is thinking that IHC diagnosis of ATRX in tumor is a really important issue for the Telomere maintenance mechanism study in malignant tumors.

>At this moment, we will continuously perform both IHC and molecular test for ALT, and correlate the results with clinical behavior of neuroblastomas.

Although authors did the above comments to my question, they haven’t reported the exact relation between the ATRX IHC and molecular test for ALT. I hope they will report the exact relation in the not so distant future.

  1. About TERT expression, MYCN (n-MYC) is known to induce TERT transcription. Authors can mention the TERT expression in the MYC subgroup by immunohistochemistry?

Response: In our UH NB subgrouping, if we detect TERT as well as MYC family proteins, these cases are considered to be MYC subgroup, whereas TERT high and MYC/MYCN negative cases are considered to be TERT subgroup. In fact, in our preliminary study shows that there is a clear separation between cases with TERT high and MYC/MYCN negative and cases with TERT high and MYC/MYCN high.

Reviewer 1:

> In fact, in our preliminary study shows that there is a clear separation between cases with TERT high and MYC/MYCN negative and cases with TERT high and MYC/MYCN high.

Again I hope they will report their findings of TERT and MYC/MYCN expression in the not so distant future.

  1. Page 10 of 26, 4.2. TERT subgroup, authors described about TERT promoter hypermethylation in refs 36/37, however, ref 36 is about medulloblastoma and relationship between TERT expression and TERT promoter was not written in ref 37.

Response: This sentence simply suggests a possibility of TERT promoter hypermethylation being one of the mechanisms for TERT overexpression in NB. That was the reason why it was stated in the manuscript: “TERT promoter hyper-methylation may also be involved in its activating mechanism“ based on refs 36 and 37. The ref 36 shows that TERT promoter hypermethylations associated with TERT overexpression in other cancer types are clearly present in a subset of NB cases, and in medulloblastoma, which exhibit a very similar biological characteristic to neuroblastoma, TERT promoter hypermethylation is associated with TERT overexpression (ref 37). Therefore, we added a sentence: “This possibility should be explored in neuroblastoma further” to avoid misinterpretation of the sentence. Please also note that refs 36 and 37 are now 41 and 42.

Reviewer 1: The sentence which they added is good for readers’ understanding.

  1. Recently, a Science Trans Medicine paper “ALT neuroblastoma chemoresistance due to telomere dysfunction–induced ATM activation is reversible with ATM inhibitor AZD0156” was published. This paper is important for this area and authors had better mention in this review.

Response: The article is now discussed and included in the MS [27]. We also modified the section 5.2.2 “Inhibitors for ALT phenotype” to discuss the recent finding on potential therapeutic approaches against ALT+ neuroblastoma (yellow highlighted).

Reviewer 1: These are OK.

Minor points:

  1. Page 7 of 26, lines 137-138, [17] can be moved to the last of this sentence?

Response: We have moved [17] to the end of sentence.

Reviewer 1: OK.

  1. Page 16 of 26, lines 574-576 are OK?

Response: The problem was fixed. Also note that this section has been extensively modified now based on the suggestions from the editorial staff and reviewers.

Reviewer 1: OK.

  1. Page 16 of 26, line 578, What is the “8”?

Response: This was a typo and was removed.

Reviewer 1: OK.

Reviewer 3 Report

It's appropiate to be publish.